# Digital economy, innovation factor allocation and industrial structure transformation—A case study of the Yangtze River Delta city cluster in China

**Xinfeng Chang**[1]**, Zihe Yang**[1]**\***, **Abdullah**[2]

**1** School of Finance and Economics, Jiangsu University, Zhenjiang, 212013, China, **2** Pakistan Air Force Karachi Institute of Economics and Technology, College of Management Sciences, Karachi, Pakistan

\* 2212119018@stmail.ujs.edu.cn

**Data Availability Statement:** All relevant data are within the manuscript and its Supporting Information files.

## Abstract

The attainment of regional high-quality development necessitates the critical role of the digital economy in facilitating the transformation of industrial structures. This study intends to investigate the effect of the digital economy on industrial structure transformation from the perspective of innovation factor allocation using a panel dataset of 41 cities in the Yangtze River Delta region for the period from 2011 to 2020. This paper considers four dimensions to measure the level of industrial structure transformation i.e. industrial structure servitization, industrial structure upgradation, service industry structure upgradation and industrial interaction level. The results of the study suggest that the digital economy can significantly improve industrial structure transformation. The results remain consistent even after several robustness checks. Further, the analysis of the mechanism of action shows that the digital economy can promote industrial structure transformation by optimizing the innovation factor allocation. The study provides several policy implications for the digital economy and its role in the promotion of industrial structure transformation.

## 1. Introduction

At present, China's economy is at an important node in the transition from high-speed growth to high-quality development. In the new stage, China is facing multiple challenges from home and abroad, such as the impact of the COVID19 and anti-globalization. The contradiction between the traditional low-end locked industrial structure and the new stage is becoming increasingly prominent. Therefore, it is the only way for China's high-quality economic development to continuously promote the evolution of the economy from factor-driven to innovation-driven and promote the balanced optimization of industrial structure. At the same time, the economies around the globe are transforming from traditional to digital economies that use artificial intelligence, blockchain, cloud computing and big data as a new economic model. The change has penetrated into all aspects of the economy from production to sales [1]. Digital economy can promote the flow of resource elements, reduce resource mismatches, enable the upgrading of traditional production factors. According to the data, the

**Funding:** This research was supported by the College Student Scientific Research Project of Jiangsu University (Project No. 22C089).

scale of China 's digital economy will reach 50.2 trillion yuan in 2022, and the proportion of digital economy in GDP will reach 41.5%. Digital economy is gradually becoming a key force driving a new round of scientific and technological revolution and industrial transformation. Further, it is worth noting that the Yangtze River Delta city cluster is the only world-class city cluster in China. It is a metropolitan area covering 41 cities, with a GDP of $427 million, an area of 350,000 square kilometers and a population of 235 million. The dataset from the Yangtze River Delta cities seems adequate for this study as it has a high degree of openness and strong innovation ability.

Industrial structure transformation remains an interesting domain for the researchers and academicians. The extant literature on industrial structure transformation can be categorized into three main domains i.e. connotation, measurement and the driving factors. Previous studies have used the theory of economic growth stages proposed by Austrian economist Walte that has laid its theoretical foundation. Scholars suggest that industrial structure transformation is an important tool to promote economic transformation [2]. Several studies argue that the transformation of industrial structure is the evolution process that moves from low to high added value [3,4]. Further, the measurement of industrial structure transformation domain can be divided into two categories i.e. single index measurement and multi-index measurement. The single index measurement includes industrial structure advance coefficient, Moore value, industrial structure level coefficient, output value ratio, the proportion of output value of each industry and the sum of product of productivity [5,6]. Moreover, the multi-index measurement measures the transformation of industrial structure from two aspects i.e. rationalization and advancement [7]. Portes and Evans [4] used the degree of industrial structure optimisation and the speed of industrial structure transition to measure. Ganadded the perspective of industrial integration development in the measurement [8]. Moreover, past studies suggested different factors that contribute to the industrial structure transformation such as industrial policies [6], capital investment [9], technological innovation [10], and financial development [11].

The digital economy and its effect on the industrial structure transformation has gained the attention of many researchers recently. Past studies have explored the internal mechanism of a digital economy and its effect on industrial structure transformation from several paths. First, several scholars argue that the digital economy can significantly promote the transformation of industrial structure. For example, digital economy has a positive spatial spillover effect on upgradation of the industrial structure[12,13]. Second, scholars struggled to find how digital economy promote industrial structure transformation and suggest that urbanization [14], technology innovation[15], labor efficiency (Wu, 2022) and factor allocation [16] help in industrial structure transformation.

In summary, the existing literature has conducted in-depth research on digital economy and industrial structure transformation from different perspectives, but there are still the following limitations: (1) the existing literature focus on how digital economy affect the industrial structure transformation. However, few studies have discussed how internal mechanism of digital economy helps in transforming the industrial structure from the perspective of innovation factor allocation. (2) Majority of the past studies have considered macro factors for the measurement of industrial structure transformation i.e. the ratio of the added value of the tertiary industry to the added value of the secondary industry, while ignoring the impact of labor within the industry, industrial integration and lacks systematic research on upgradation of the industrial structure. (3) It is worth noting that in China 's development plan, the city and county levels are more instructive in the specific implementation of the policy [17]. Previous studies have provided evidence from the perspective of province, few studies have been conducted from the level of prefecture-level cities to grasp the intra-regional connections.

By analyzing the shortcomings of the existing literature, this paper focuses on the effect and mechanism of digital economy on industrial transformation from the perspective of innovation factor allocation. The contributions of this paper are: (1) We incorporate digital economy, innovation factor allocation and industrial structure transformation in a single framework. We have considered digital economy as the basis of industrial transformation and analyzed the intermediary role of innovation factor allocation in the promotion of industrial structure transformation by digital economy. (2) we extended the literature by using a comprehensive measurement of industrial structure transformation that considered its four dimensions which are industrial structure servitization, industrial upgradation, service industry structure upgradation and industrial interaction level. (3) This paper extends the existing literature by providing evidence from the perspective of the Yangtze River Delta city where the digital economy is developing rapidly. The rest of the article is structured as follows: section 2 highlights the theoretical analysis and research hypothesis, while sections 3 and 4 deal with methods and data, and results and discussion, respectively. Lastly, section 5 presents the conclusion and recommendations.

## 2. Theoretical analysis and research hypothesis

Some studies have found that although the evolution of the economic structure measured by increasing the proportion of the service industry shows that China's economy is moving towards a higher level, the structural problems within the industry have caused the benign interaction of the industry to be hindered [18]. This kind of economic servitization will make the economy face the risk of moving from the real to the virtual, and there will be excessive servitization. Based on the background of high-quality development, this paper defines the transformation of industrial structure: The change in the proportional relationship between macro industries based on the optimization of factor composition and ratio within microenterprises and the upgrading of technology and products within meso-industries is the endogenous basis of industrial structure transformation. This paper will explain the impact of the digital economy on the transformation of industrial structure from the four dimensions of industrial structure servitization, industrial structure upgradation, service industry structure upgradation, and industrial interaction level, and then put forward the corresponding research hypothesis.

### 2.1. The direct effect of the digital economy on the transformation of industrial structure

The impact of the digital economy on the transformation of industrial structure is visible as it promotes the servitization of industrial structure, the upgradation of industrial structure, the upgradation of service industry structure, and the level of industrial interaction. The digital economy reduces transaction cost and transform the servitization of industrial structures. The digital economy penetrates through digital technology and reduces the information barriers between enterprises and industries. Further, it promotes industrial efficiency and create synergy while using modern science and technology [19]. It innovates the production process of traditional industries through innovative management options and realizes the servitization of industrial structure [20]. Further, digital economy is profoundly changing all interrelated value-added links within the product lifecycle of the manufacturing industry and promotes transformation and upgradation of the manufacturing enterprises [21]. It helps in immediate gathering and disseminating important information that may help in efficient decision-making through different digital technologies such as big data and the internet of things and have overcome several issues of the labors in the manufacturing industry [22]. Digital economy also

promote the transformation and upgradation of traditional industrial enterprises [23]. Moreover, the digital economy supports the rapid development of new forms of knowledge-intensive services, which helps to upgrade the structure of the service industry. The development of the digital economy has brought convenience to data integration, resource flow, and value sharing. The wide application of digital technology in the service industry has greatly promoted the rapid development of new forms of knowledge-intensive services such as integrated offices, online medical care, online education, and cross-border services. The continuous advancement in the digital technology is perhaps fulfilling the modern requirements which are important in upgradation of service industry structure and for the improvement in the labor productivity [24]. Finally, the digital economy promotes organizational change and helps in gathering important information in lesser time. The digital economy can promote interaction among different industries using the available information on the internet and through digital inclusive finance. The digital technology continues to change the production and organizational methods in various industries. Further, this will reduce the traditional barriers and help industries to integrate leading to sustainable development [25]. Based on the above discussion, we develop the following hypothesis.

**Hypothesis 1:** The digital economy has a direct role in promoting the transformation of industrial structures.

## 2.2. The mediating effect of innovation factor allocation in digital economy promoting industrial structure transformation

Innovation is the key to the transformation of industrial structures. From the perspective of human factor, the digital economy can indirectly promote the transformation of the industrial structure by improving the innovation in elements such as humans, knowledge, technology, and systems. The construction of human organizations may become the source of innovation research [26]. At the same time, the labor force engaged in R&D activities has high education and strong skills [27], which can provide intellectual support for the transformation of industrial structure. The implementation of the internet and modern communication technology, the integration of data elements and labor force can promote the allocation efficiency of human innovation elements. Thus, innovation in the human factor may help them to perform creative activities which results in transformation of industrial structure.

From the perspective of innovation in the knowledge elements, the digital economy can use modern digital technologies to swiftly screen out information that is conducive to innovation, improve the capacity of information storage and use it innovative ways to maximize value and drive industries to higher level maximize value [28,29]. From the perspective of technological innovation elements, the development of the digital economy has accelerated the speed of information transmission. The digital economy may be helpful in build an information-sharing platform, integrate internal and external resources of enterprises which will create opportunities for collaboration and result into transparent environment. This efficiency in the technological innovation may improve the environment of enterprise and upgrade the industrial structure. From the perspective of system innovation elements, with the development and expansion of the digital economy, the country has gradually attached importance to institutional innovation in digital platforms, data security, and artificial intelligence. The improvement in the institutional innovation can motivate the innovative behavior among employees that will smoothly transform the industrial culture and bring technological innovation in the industry [30]. Based on the above discussion, we develop the following hypothesis:

**Hypothesis 2**: The digital economy can indirectly promote the transformation of the industrial structure by improving the level of innovation factor allocation.

## 3. Models, variables and data

### 3.1. Model setting

**3.1.1. Benchmark model construction.**   In order to explore the direct impact of digital economy on the transformation of industrial structure, the following benchmark regression model is constructed:

$$Y_{i,t} = \alpha_0 + \alpha_1 Dig_{i,t} + \alpha_c Z_{i,t} + \mu_i + \delta_t + \varepsilon_{i,t} \tag{1}$$

Where, $i$ and $t$ represent the city and time respectively, and $Y_{i,t}$ is the industrial structure transformation of the explained variable, which is measured from the four dimensions i.e. of industrial structure servitization, industrial structure upgradation, service industry structure upgradation and industrial interaction level. $Dig_{i,t}$ represents the development level of digital economy while, vector $Z_{i,t}$ represents a series of control variables. Further, $\mu_i$ represents the individual fixed effect that the city does not change with time, $\delta_t$ represents the time fixed effect, and $\varepsilon_{i,t}$ is the random disturbance term.

**3.1.2. Mediating effect model.**   The following model was developed to test whether innovation factor allocation mediates the relationship between digital economy and industrial structure transformation consistent with Wen and Ye [31].

$$M_{i,t} = \beta_0 + \beta_1 lnDig_{i,t} + \beta_c Z_{i,t} + \mu_i + \delta_t + \varepsilon_{i,t} \tag{2}$$

$$Y_{i,t} = \gamma_0 + \gamma_1 Dig_{i,t} + \gamma_2 M_{i,t} + \gamma_c Z_{i,t} + \mu_i + \delta_t + \varepsilon_{i,t} \tag{3}$$

Where, $M_{i,t}$ is the mediating variable i.e. innovation factor allocation, and the premise of the mediating effect is the significance of $\alpha_1$ in Eq (1) which indicates that the digital economy has an impact on the transformation of industrial structure. If the coefficients $\beta_1$ and $\gamma_2$ are significant, the indirect effect exists. If any of them is not significant then the Sobel test is performed. If the hypothesis of $\gamma_2 = 0$ is significantly rejected, the mediating effect exists, and vice versa.

### 3.2. Variable measure and description

**3.2.1. Explanatory variables: Industrial structure transformation.**   The transformation of industrial structure refers to the improvement of production efficiency and the transfer of production factors from low-efficiency industries to high-efficiency industries. It is usually measured by the upgradation and the rationalization of industrial structure. On one hand, the digital economy shifts the factors of production from low-efficiency industries to high-efficiency industries, which promotes the improvement of production efficiency. On the other hand, it makes the development trend of mutual integration and blurred boundaries between industries more effectively. At the same time, the industrial policy that intend to enhance service industry proportion may misallocate resources between industries which results in excessive servitization. It is impossible to accurately analyze the transformation of the industrial structure by using one proxy i.e. upgradation and rationalization of the industrial structure. Therefore, this paper measures it from the four dimensions to accurately measure it i.e. industrial structure servitization, industrial structure upgradation, service industry structure upgradation, and industrial interaction level, as shown in Table 1.

1. Industrial structure servitization (Insev) In this paper, the servitization of industrial structure is used to describe the transformation of industrial structure at the macro level. It represents the changes in the proportion of the three industries at the macro level. It is measured by the ratio of the added value of the tertiary industry to the secondary industry,

Table 1. Industrial structure transformation level measurements.

| First-level indicators | Second-level indicators | Third-level indicators | index attribute |
|---|---|---|---|
| Industrial structure transformation | Industrial structure servitization | The added value of the tertiary industry / the added value of the secondary industry | + |
| | Industrial structure upgradation | Total regional industrial profits and taxes | + |
| | Service industry structure upgradation | Producer services added value / tertiary industry added value | + |
| | Industrial interaction level | Herfindahl index | + |

and combined with the other three indicators to comprehensively evaluate the transformation of industrial structure.

2. Industrial structure upgradation (Manh): The increase in the manufacturing value chain or the transformation and upgradation from traditional to advanced manufacturing industry will be reflected through the increase in the value chain. Therefore, this study uses the total regional industrial profits and taxes to measure the advanced industrial structure. To a certain extent, this index reflects the value added in the manufacturing industry. It is believed that the total profit and tax of the high-tech industry is usually higher as compared to others. If the total amount of industrial profits and taxes in a region is higher, it shows that the industrial level in the region is also higher.

3. Service industry structure upgradation (Sevh): The upgradation of service industry structure shows the rapid development of emerging industries and producer services compared to traditional industries. Producer services can not only effectively overcome the Baumol's disease because of its high productivity, but also support the development of advanced manufacturing industry. This paper uses the ratio of employment in producer services to employment in the tertiary industry to measure the structure of the service industry [32]. According to the classification criteria of the National Bureau of Statistics (2005), producer services mainly cover transportation, warehousing and postal services, financial services, leasing and business services, scientific research, technical services and geological exploration, information transmission, computer services and software industries.

4. Industrial interaction level (Indi): In the modern era, industrial interaction and integration is an effective development model that improves productivity and competitiveness. Industrial interactive integration refers to the process of removing the barriers in the industry for industrial growth. This cross-industry interaction enhances technology innovation which enhances industrial agglomeration and industrial interactive integration. In the era of service economy and digital economy, service products will be put into the economic production activities of various industrial sectors as intermediate products on a large scale. Therefore, the level of industrial interaction is one of the important characteristics of industrial structure transformation. In this study, Herfindahl index (HHI) is used to represent, $Q_i$ represents the output value of the $i$ industry, $Q$ represents the regional GDP which is presented in the following model:

$$HHI = \sum_{i=1}^{N} \left( \frac{Q_i}{Q} \right)^2 \tag{4}$$

**3.2.2. Core explanatory variables: Digital economy development level.** This study measures the digital economy at the prefecture-level city level. The data has been collected based on its availability [33]. Kapur and Kesavan [34]stated that, when the data source produces a

low-entropy value, the event carries more "information". The entropy method is an objective and comprehensive weighting method, which is based on the dispersion degree of the evaluation index data to measure the index weight, so we use the entropy method to calculate the digital economy considering the level of development of internet and digital inclusive finance. Further, the level of Internet development is divided into four three-level indicators: first, the output level of internet-related industries measured by the total amount of telecommunications business; second, the internet-related industry practitioners with the number of computer services and software industry practitioners to characterize; third, the internet penetration rate expressed by the number of Internet broadband access users in 100 people; the fourth is the mobile phone penetration rate expressed as the number of mobile phone users per 100 people. The development level of digital inclusive finance is characterized by the digital inclusive finance index compiled by the Digital Finance Research Center of Peking University and Ant Financial Services Group. The principal component analysis method is used to standardize the relevant data and reduce the dimension, so as to obtain the development level of the digital economy at the city level.

**3.2.3. Mediating variable: The level of innovation factor allocation.** We constructed a comprehensive index for the allocation of innovation elements comprising four dimensions i.e. human, knowledge, technological and institutional innovation. We measured human innovation by the full-time personnel for R&D while the number of colleges and universities were used to measure the human organizations. Further, for the knowledge innovation several aspects are considered such as knowledge retention, technological innovation, quality of technological innovation. The internal R&D expenditure was used to measure the knowledge retention and the quantity of invention patents were used to measure the quality of technological innovation. Similarly, the sum of the quantity of utility model patents and design patents is used to measure the quantity of technological innovation. Moreover, we measured the institutional innovation factor by the total collection of books in the public libraries and government's expenditures on science and education. As per the objective weighting method, the entropy TOPSIS is used to calculate the level of innovation factor allocation (Inf). The system of measurement indicators is shown in Table 2.

**3.2.4. Control variables.** To study the effect and mechanism of the digital economy on the transformation of industrial structure more comprehensively, this paper refers to the existing literature [11,35–39] and used several control variables such as economic development degree, infrastructure level, financial development level, government intervention and population density. Economic development degree (Pgdp) refers to the degree of economic development which is an important driving force to promote the upgradation of China's industrial structure. Infrastructure level (Bins) is important for the economic development and is the basis for the transformation of industrial structure. Financial development level (Fin) which is also required for the industrial development and this problem can be solved by the funds

**Table 2. Evaluation index system of innovation factor allocation level.**

| First-level indicators | Second-level indicators | Third-level indicators | index attribute |
|---|---|---|---|
| innovation factor allocation | Human innovation elements | Number of ordinary colleges and universities | + |
| | | R&D personnel full-time equivalent | + |
| | Knowledge innovation elements | R&D internal expenditure | + |
| | Technological innovation elements | Amount of invention patents obtained | + |
| | | The amount of utility model patents and design patents obtained | + |
| | Institutional innovation elements | The total collection of books in public libraries | + |
| | | Financial expenditure on science and education | + |

available in the capital markets. We used the proportion of loan balance of financial institutions to the GDP for measuring the level of financial development. Government intervention (Gov) also impact the industrial structure and government plays an important role of the regulator. Lastly, Population density (Lnpop) is used as a control variable which is an important factor that affects the transformation of industrial structure. It is measured by the ratio of regional resident population to urban land area.

### 3.3. Data sources and descriptive statistics

This paper used a balanced panel dataset comprising 41 cities in the Yangtze River Delta for the period 2011 to 2020. The data has been extracted from "Jiangsu Statistical Yearbook ", "Zhejiang Statistical Yearbook", "Anhui Statistical Yearbook ", "China City Statistical Yearbook ", each city Statistical Yearbook, and the EPS database. Some missing data was supplemented by the linear interpolation method. Table 3 presents the descriptive statistics of all the variables used in this study.

Table 3 suggests that the standard deviation of industrial structure servitization, service industry structure upgradation, and industrial interaction level is small which implies that the data fluctuation is not large. Further, the minimum and maximum values of industrial structure upgradation are significantly different, indicating that the development gap of industrial structure among cities in the Yangtze River Delta region is large. Moreover, the descriptive statistics suggests that the mean value of digital economy is small but the standard deviation is large. This finding is consistent with [12]. Similarly, the allocation level of innovation elements also has a similar mean and standard deviation. The descriptive statistics of control variables suggest significant differences in economic development, infrastructure level, financial development level, government size, and population density among different cities.

## 4. Empirical results and analysis

### 4.1. Benchmark regression analysis

We perform several diagnostic checks to ensure that the dataset meets the basic assumptions of regression. First, we perform a 1% tail reduction on the panel data in order to remove outliers. Second, we use Hausman test to check whether fixed or random effect model is appropriate, and the results support the use of the fixed effect model. In the actual regression, the time trend of industrial structure servitization and service industry structure upgradation is not

**Table 3. Descriptive statistics of variables.**

| Type | Name | Observations | Mean | Standard deviation | Minimum value | Maximum value |
|---|---|---|---|---|---|---|
| Explained variable | Industrial structure servitization | 410 | 0.986 | 0.340 | 0.313 | 2.751 |
| | Industrial structure upgradation | 410 | 12.604 | 2.046 | 7.676 | 17.696 |
| | Service industry structure upgradation | 410 | 0.273 | 0.086 | 0.073 | 1.000 |
| | The level of industrial interaction | 410 | 0.444 | 0.048 | 0.342 | 0.613 |
| Core explanatory variables | The development level of digital economy | 410 | 0.000 | 0.470 | -0.800 | 2.430 |
| Control variables | Level of economic development | 410 | 17.102 | 0.956 | 15.131 | 19.774 |
| | Infrastructure level | 410 | 22.697 | 7.192 | 4.040 | 46.400 |
| | Degree of financial development | 410 | 1.181 | 0.437 | 0.472 | 3.054 |
| | Government intervention | 410 | 0.010 | 0.003 | 0.005 | 0.023 |
| | Population density | 410 | 11.394 | 1.222 | 8.396 | 14.548 |
| Mediator variable | The level of innovation factor allocation | 410 | 0.089 | 0.126 | 0.004 | 0.839 |

**Table 4. Benchmark regression results.**

| Variable | (1) | (2) | (3) | (4) | (5) | (6) | (7) | (8) |
|---|---|---|---|---|---|---|---|---|
| | Insev | Insev | Servh | Servh | Manh | Manh | Indi | Indi |
| Dig | 0.5507*** | 0.2956*** | 0.0356*** | 0.0829*** | 0.5105*** | 0.4413*** | 0.0583*** | 0.0557*** |
| | (0.0194) | (0.0626) | (0.0067) | (0.0221) | (0.1206) | (0.1227) | (0.0123) | (0.0115) |
| Pgdp | | 0.2206 | | -0.0650*** | | -0.2818*** | | 0.0494*** |
| | | (0.0737) | | (0.0260) | | (0.1211) | | (0.0114) |
| Bins | | 0.0117*** | | -0.0008 | | -0.0012 | | -0.0006 |
| | | (0.0030) | | (0.0011) | | (0.0044) | | (0.0004) |
| Fin | | -0.0144 | | 0.0346* | | -0.2498*** | | -0.0180** |
| | | (0.5722) | | (0.0202) | | (0.0894) | | (0.0084) |
| Fdi | | -2.5948 | | 0.0358** | | 0.3363*** | | 0.0390*** |
| | | (4.3853) | | (1.5497) | | (0.0645) | | (0.6070) |
| lnpop | | 0.1274*** | | -0.0138 | | 0.1124*** | | -0.0440*** |
| | | (0.0374) | | (0.0135) | | (0.0543) | | (0.0051) |
| Cons | 0.9881*** | -3.8263*** | 0.2709*** | 1.4119*** | 12.7735*** | 16.6718*** | 0.4731*** | -0.0837 |
| | (0.0059) | (1.3130) | (0.0020) | (0.4640) | (0.0702) | (2.1613) | (0.0072) | (0.2034) |
| N | 410 | 410 | 410 | 410 | 410 | 410 | 410 | 410 |
| $R^2$ | 0.6152 | 0.7154 | 0.0708 | 0.1221 | 0.2012 | 0.3105 | 0.1319 | 0.3659 |
| Time fixed | NO | NO | NO | NO | YES | YES | YES | YES |
| City fixed | YES | YES | YES | YES | YES | YES | YES | YES |

Note: * * * *, * * and * indicate significant at the 1%, 5% and 10% levels, respectively; the values in brackets represent robust standard errors, the same below.

obvious, so the one-way fixed effect model without time effect is chosen for the final regression of the two dimensions.

Table 4 reports the benchmark regression results of industrial structure transformation driven by the digital economy. The results of columns 1, 3, 5 and 7 show that the development of the digital economy has significantly promoted the servitization of industrial structure, the upgradation of industrial structure, the upgradation of service industry structure and the level of industrial interaction without considering the control variables. Further columns 2, 4, 6, and 8 report results after inclusion of control variables. The results are consistent which suggest that considering the differences in the degree of economic development and infrastructure construction in different cities, the development of a digital economy can significantly promote the transformation of industrial structure. It is also found that the promotion effect of the digital economy on industrial structure servitization and industrial structure upgradation is much greater than that on service industrial structure upgradation and industrial interaction level. Overall, the results suggest that the digital economy may enable transformation of industrial structure, which not only promote the servitization of industrial structure at the macro level but also effectively promote the integration of service industry and agriculture which optimize the industrial internal structure, promote the development of high-tech industry, and produce productive service industries. The digital economy may serve as a strong driving force which promote the evolution of industrial structure and industrial internal structure to the middle and high end. Therefore, the development of the digital economy in the Yangtze River Delta region enables the servitization of advanced industrial structure instead of "real-to-virtual" transformation of the economy. Hence, we find support for H1.

Further, the results suggest that the degree of economic development is conducive to improving the level of industrial structure upgradation and industrial interaction level, which is consistent with scholars [3,40]. However, the impact of the industrial structure upgradation

and the upgradation of the service industry structure is significantly negative, which suggest that China suffers from insufficient supply of high-quality products and the dependence on foreign products. The positive coefficient of infrastructure construction for the servitization of industrial structure indicates that infrastructure construction can improve the cost of factor flow which promotes the digital economy, and transform the industrial structure to servitization. The level of financial development also promotes the upgradation of the service industry structure and new products are developed using internet and finance that has significantly improved the industry. However, the demand of finance from industrial upgradation does not match the supply from financial institutions which may be a hindrance towards industrial structure upgradation. The population density promotes the service of industrial structure and the advancement of industrial structure, which indicates that the development of a digital economy can attract a large number of high-tech talents which helps in transforming traditional industry to tertiary industry. However, the increase in population will shift the focus of labor to low-end service industries which may become the hindrance in the development of the service industry.

## 4.2. Mediation effect test

This study analyzes the transmission path of digital economy development to industrial structure transformation. Table 5 reports the regression results of the intermediary effect model. Column 1 and 4 present the results that test the impact of the digital economy on the innovation factors allocation without controlling the time effect and with controlling the time effect, respectively. The significant coefficients indicate that the development of digital economy can significantly improve the innovation factors allocation level. As the economy started transition into digitalization, the transaction cost of innovation factors is reduced which will promote the flow of factors such as talents and knowledge which efficiently allocate innovation factors.

**Table 5. Regression results of innovation factor allocation transmission mechanism.**

| Variable | (1) | (2) | (3) | (4) | (5) | (6) |
|---|---|---|---|---|---|---|
| | Inf | Insev | Servh | Inf | Manh | Indi |
| Dig | 0.0795*** | 0.2628*** | 0.0841*** | 0.1913*** | 0.4237*** | 0.0453*** |
| | (0.0192) | (0.0637) | (0.0226) | (0.0257) | (0.1321) | (0.0237) |
| Inf | | 0.4122*** | -0.0147 | | 0.0919 | 0.0546*** |
| | | (0.1702) | (0.0606) | | (0.2536) | (0.0237) |
| Pgdp | -0.0253 | 0.2310*** | -0.0654*** | -0.0055 | -0.2813*** | 0.0497*** |
| | (0.0256) | (0.0715) | (0.0261) | (0.0254) | (0.1213) | (0.0113) |
| Bins | -0.0014 | 0.0122*** | -0.0008 | 0.0005 | -0.0013 | -0.0007 |
| | (0.0009) | (0.0030) | (0.0011) | (0.0009) | (0.0044) | (0.0004) |
| Fin | 0.0092 | -0.0183 | 0.0347** | -0.0293 | -0.2471*** | -0.0164** |
| | (0.0175) | (0.0552) | (0.0203) | (0.0188) | (0.0898) | (0.0084) |
| Fdi | 0.0265** | -0.0369 | 0.0361** | 0.0483*** | 0.3318*** | 0.0364*** |
| | (0.1343) | (0.0438) | (0.0156) | (0.0135) | (0.0657) | (0.6141) |
| lnPop | 0.0507*** | 0.1065*** | -0.0013 | 0.0397*** | 0.1087** | -0.0458*** |
| | (0.0117) | (0.0389) | (0.0134) | (0.0114) | (0.0055) | (0.0053) |
| Cons | 0.1881*** | -3.9038*** | 1.4146** | -0.0071*** | 16.6725*** | -0.0833 |
| | (0.4022) | (1.3047) | (0.4647) | (0.4535) | (2.1639) | (0.2021) |
| Sobel test value | | 0.0350*** | -0.0095*** | | -0.1950** | 0.3294*** |
| N | 410 | 410 | 410 | 410 | 410 | 410 |
| R$^2$ | 0.3101 | 0.7200 | 0.1240 | 0.8460 | 0.7187 | 0.3754 |
| Time fixed | NO | NO | NO | YES | YES | YES |
| City fixed | YES | YES | YES | YES | YES | YES |

The columns 2, 3, 5, and 6 present the regression results of the benchmark regression plus the mediating variables. It is found that in columns 2 and 6, the coefficients of the development level of the digital economy and the allocation of innovation factors are significantly positive, and the absolute value of the coefficient of the development level of the digital economy is reduced compared with the benchmark regression, indicating that the digital economy plays a significant mediating role in driving the servitization of industrial structure and the level of industrial interaction. Further, the innovation factors allocation does not have any significant impact on the upgradation of the service industry structure and the industrial structure in column 3 and 5. However, after the Sobel's test, the hypothesis that there is no mediating effect is significantly rejected which implies that the digital economy indirectly affects the service structure upgradation and the industrial structure upgradation by promoting the innovation factors allocation. The development of digital economy stimulates the demand for innovative factors, and expands the supply scale of innovative factors through the informatization and digitization of enabling factors, which lays a factor foundation for industrial transformation. At the same time, the innovation of participation mode accelerates product innovation, business integration and high-end upgrading. Hence, we find support for H2.

### 4.3. Endogeneity and robustness test

**4.3.1. Treatment of endogenous problems.** In the above models, there are two possible endogenous problems: First, there may be a two-way causal relationship between the digital economy and industrial structure transformation as the increase in technological demand for industrial structure transformation may in turn affect the development of the digital economy. Second, there may be several variables which are omitted that can make results bias although the impact of control variables such as economic development level and financial development level is considered. To overcome these possible endogenous problems, we have used an instrumental variable approach to estimate the model. By constructing the interaction term between the number of fixed telephones per 100 people in 1984 and the number of urban internet users in the previous year as the instrumental variable of the digital economy development level. We used the two-stage least squares method for analysis and the results are presented in Table 6. The first-stage regression results show that there is a significant positive correlation between the instrumental variables and the digital economy. The rationality test results for instrumental variables show that the p-value corresponding to the Kleibergen-Paaprk LM statistic is less than 0.01, which significantly rejects the null hypothesis that "insufficient identification of instrumental variables" at the 1% level. Further, the Kleibergen-Paapr-Wald F test value is 37.398 which is greater than the critical value of 16.38 at the 10% level of the Stock Yogo test passing the weak tool test which indicates that the instrumental variables selected in this paper are reasonable. Moreover, the second stage regression results show that the development level of the digital economy still promotes the servitization of industrial structure, the advancement of industrial structure, the advancement of service industry structure, and the level of industrial interaction. Both are significant at the 1% level, which is consistent with the main regression results.

**4.3.2. Stability test.** As discussed earlier, we have measured the explanatory variables i.e. industrial structure transformation from four dimensions. These four dimensions are complementary, and the regression results between them can explain the promotion effect of the digital economy on industrial structure transformation to a certain extent. In addition, we use the alternate variable measurement, excluding municipalities and provincial capitals to further validate the robustness of the results.

(1) Replacement variable measure method

**Table 6. Regression results of instrumental variable test.**

| Variable | (1) | (2) | (3) | (4) |
|---|---|---|---|---|
| | Insev | Manh | Servh | Indi |
| Dig | 0.7754*** | 0.8278*** | 0.0480*** | 0.2104*** |
| | (0.2078) | (0.2750) | (0.0174) | (0.0425) |
| Control variable | YES | YES | YES | YES |
| Cons | 6.1049 | 26.5451*** | 0.6781 | 4.3051*** |
| | (4.2696) | (5.6486) | (0.4804) | (0.8723) |
| Kleibergen-Paaprk WaldF | 37.3979 {16.38} | 37.3979 {16.38} | 37.3979 {16.38} | 37.3979 {16.38} |
| Kleibergen-PaaprkLM | 10.127 [0.0015] | 10.127 [0.0015] | 10.127 [0.0015] | 10.127 [0.0015] |
| N | 410 | 410 | 410 | 410 |
| $R^2$ | 0.8757 | 0.9621 | 0.0911 | 0.7433 |
| Time fixed | YES | YES | YES | YES |
| City fixed | YES | YES | YES | YES |
| The first stage regression result dependent variable:Dig | | | | |
| IV | 0.0465*** | 0.0465*** | 0.0460*** | 0.0465*** |
| | (0.0076) | (0.0076) | (0.0064) | (0.0076) |

In order to eliminate the interference of the variable measurement method with the estimation results, this paper uses the entropy method to measure the digital economic development index and re-estimate the model and the results after replacement are presented in Table 7. The coefficient sign and significance of the digital economy development level are consistent with the main results which implies that the results are robust to several measurements of digital economy.

(2) Delete provincial capitals and municipalities

The sample data in this paper includes 41 cities of the Yangtze River Delta region.

Since, the level of each city in terms of economy, finance and policy is different from other therefore, the regression results are also different. In this section, we exclude four cities from our sample and re-estimated our models. The results are presented in Table 8 which suggest that the results are consistent with our main results which imply that the digital economy has a significant positive effect on the transformation of industrial structure.

## 5. Conclusions and recommendations

This study intends to investigate the effect of the digital economy on industrial structure transformation from the perspective of innovation factor allocation using panel dataset of 41 cities

**Table 7. Replacement variable measure method regression results.**

| Variable | (1) | (2) | (3) | (4) |
|---|---|---|---|---|
| | Insev | Manf | Servh | Indi |
| Dig2 | 1.4859*** | 0.8900** | 0.1402* | 0.1885*** |
| | (0.2280) | (0.3500) | (0.0743) | (0.0266) |
| Control variable | YES | YES | YES | YES |
| Cons | 0.8260*** | 15.0172*** | 0.0582*** | -0.3453* |
| | (1.3771) | (2.1141) | (0.2353) | (0.1902) |
| N | 410 | 410 | 410 | 410 |
| $R^2$ | 0.7784 | 0.2982 | 0.0988 | 0.4081 |
| Time fixed | YES | YES | YES | YES |
| City fixed | YES | YES | YES | YES |

**Table 8. Regression results of provincial capitals and municipalities deleted.**

| Variable | (1) | (2) | (3) | (4) |
|---|---|---|---|---|
|  | Insev | Manh | Servh | Indi |
| Dig | 0.1505** | 0.5947*** | 0.0771*** | 0.0463*** |
|  | (0.0695) | (0.1714) | (0.0241) | (0.0103) |
| Control variable | YES | YES | YES | YES |
| Cons | -6.4860*** | 15.8559 | 0.6781 | 1.2425*** |
|  | (1.3822) | (2.2322) | (0.4804) | (0.2053) |
| N | 370 | 370 | 370 | 370 |
| R² | 0.7124 | 0.3263 | 0.0911 | 0.1399 |
| Time fixed | YES | YES | YES | YES |
| City fixed | YES | YES | YES | YES |

in the Yangtze River Delta region for the period from 2011 to 2020. We have considered digital economy as the basis for industrial structure transformation and measured the industrial structure transformation comprehensively using its four dimensions i.e. industrial structure servitization, industrial structure upgradation, service industry structure upgradation and industrial interaction level. We have employed panel regression technique for ascertaining the relationship between the variables. The results suggest that digital economy has a significant impact on industrial structure transformation. It does not only promote the servitization but plays a crucial role in industrial structure upgradation, service industry structure upgradation and enhance industrial interaction. Further, we find that innovation factor allocation mediates the relationship between digital economy and industrial structure transformation. Digital economy can accelerate the flow and combination optimization of various innovative resources through digital, intelligent and networked organization, improve the efficiency of resource allocation and the coordinated development of industrial structure, and help to build a modern industrial system with coordinated allocation of factors, intra-industry development and inter-industry deep integration. The findings are robust to different measurements and several estimation techniques.

The results have several implications. First, the government should promote digital economy in all regions that will transform into high-quality development which will results in sustainable economic development. All regions should make a strategy to implement digital economy using artificial intelligence and blockchain which will transform the industrial structure. Second, all regions should promote innovation and incentivize firms that innovate at all levels. Government should also allocate special funds that help firms in adopting technologies that brings innovation. All regions may collaborate with each other to enhance the innovation as is it will help in industrial transformation. Third, digital economy policies may be developed according to the regional conditions. All regions should devise own strategies to enhance digital economy in order to enhance its impact on the region. Metropolitan cities may strengthen their relationship with small cities and promote innovation and digital economy which will enhance collaboration and industrial transformation. It is also suggested that the policy obstacles may be removed which restrict cities to collaborate with each other. Multi-stakeholders have an interactive effect on the choice of governance strategies, which is affected by the cost-benefit relationship of various stakeholders [41]. Local governments should assess the information technology infrastructure and make information technology investment plans accordingly to promote digital economy which will help in overall sustainable economic development and growth.

Based on the panel data from 41 cities in the Yangtze River Delta, this study takes industrial structure transformation as the explanatory variable and digital economy as the main explanatory variable and draws the key conclusion that digital economy can significantly promote industrial structure transformation. However, there are still some limitations. First, Limited to the limitations of the index data, the measurement of industrial structure upgrading in this paper is only measured from the macro inter industry and the meso industry, and the micro factors are not included in the index measurement. In the case of available data, more detailed research can be further carried out based on the level of micro enterprises. Second, limited to the limitations of index data, only representative and data-accessible indicators are selected for research in the construction of digital economy index system. With the rapid development of digital technology, the measurement index system of digital economy should keep up with the pace of development. In the future research, it is necessary to build a more perfect and more realistic digital economic evaluation index system.

## Supporting information

**S1 Data.**
(XLSX)

## Author Contributions

**Formal analysis:** Zihe Yang.

**Methodology:** Zihe Yang.

**Software:** Zihe Yang.

**Supervision:** Xinfeng Chang.

**Validation:** Zihe Yang.

**Visualization:** Zihe Yang.

**Writing – original draft:** Zihe Yang.

**Writing – review & editing:** Xinfeng Chang, Abdullah.

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
