## [Decision Letter · Decision Letter 0]

19 Feb 2024

PONE-D-23-27148Digital economy, innovation factor allocation and industrial structure transformation- A case study of the Yangtze River Delta city cluster in ChinaPLOS ONE

Dear Dr. 杨,

Thank you for submitting your manuscript to PLOS ONE. After careful consideration, we feel that it has merit but does not fully meet PLOS ONE’s publication criteria as it currently stands. Therefore, we invite you to submit a revised version of the manuscript that addresses the points raised during the review process.

We look forward to receiving your revised manuscript.

Kind regards,

Jianhua Zhu

Academic Editor

PLOS ONE

Journal Requirements:

3. We note that your Data Availability Statement is currently as follows: "All relevant data are within the manuscript and its Supporting Information files."

Reviewers' comments:

Reviewer's Responses to Questions

**Comments to the Author**

1. Is the manuscript technically sound, and do the data support the conclusions?

Reviewer #1: Yes

Reviewer #2: Yes

2. Has the statistical analysis been performed appropriately and rigorously? 

Reviewer #1: Yes

Reviewer #2: Yes

3. Have the authors made all data underlying the findings in their manuscript fully available?

Reviewer #1: Yes

Reviewer #2: Yes

4. Is the manuscript presented in an intelligible fashion and written in standard English?

Reviewer #1: Yes

Reviewer #2: Yes

5. Review Comments to the Author

Reviewer #1: The author has examined the mechanism of digital economy on industrial transformation. It is a hot topic generally. The Yangtze River delta is a very good sample area. The content is very detailed, and the theory logic is very reasonable. To improve the manuscript, I provide the following comments,

(1) Abstract is a little longer, please make it shorter and show more about the important findings to readers.

(2) in the first paragraph of Introduction, the author discussed a lot of service. It is very strange for us as a reader why you discuss service here. I can understand you when I read the measurement of industrial structure using service. My suggestion is to write more about the important of the role of digital economy in industrial structure upgrade. it wil be better not to mention some new terms (or some second-level constructs) not relevant to the title.

It has been argued that the Chinese economy is shifting from industrial economy to service economy (Sun P, 2021). The service sector of the economy is increasing rapidly as it has exceeded the proportion of manufacturing

sector. Besides its growth, the structural problem in the economy is extremely risky for its sustainability (Zheng, 2023). If the government develops one sector at the expense of other sectors, it may lead to a risk of de-realization and over-servitization (Di Meglio G, 2018).

(3) i suggest you not use Political person or political conference to prove the importance. it is better to use data, evidence, or by citing some importance references to discuss the importance. in the paper, you discuss Xi jinping, 20th party congress. All these citations are not academic style.

Mr. Xi Jinping stressed promoting the integrated development of the Yangtze River Delta. This provides a feasible path for the digital economy that may promote the transformation of industrial structure

The 20th Party Congress report proposed to "accelerate the construction of a strong manufacturing country", and the 14th Five-Year Plan will "maintain the basic stability

(4) Introduction part is too long. I suggest you reduce it into 1.5-2 pades. Please mention research gaps in literature review and give your contributions. the following framework can be adopted.

Present your topic and get the reader interested

Provide background or summarize existing research

Position your own approach

Detail your specific research problem and problem statement

your method, findings and contribution

Give an overview of the paper’s structure

(5) The Plos one is a well-established international journal. I find author cite many references writing in Chinese. Please cut them or replace them with some relevant and good journal papers.

By the way, the format should fit to Plos one style or international style like APA, Chicago and so on. the [J] style is Chinese standard, not an international one. I hope you can understand me and I did not spell them out here.

Further, I find most authors of your references are Chinese writers. Is any non-Chinese background scholars write this kind of papers? Please do the systematic review first.

I suggest you some relevant references,

on China industrial structure upgrade,

https://doi.org/10.1177/21582440221095013

on Innovation impact of industrial upgrade

http://www.verspagen.nl/index.php/bart-s-economics/29-new-perspectives-on-structural-change

https://doi.org/10.1007/s11356-022-23521-8

https://doi.org/10.1057/s41599-023-01910

on Yangtze River delta

https://doi.org/10.3389/fenvs.2022.896036

https://dx.doi.org/10.11644/KIEP.EAER.2022.26.3.411

on digital economy and industrial structure

https://doi.org/10.1371/journal.pone.0277787

https://doi.org/10.1371/journal.pone.0277259

(6)The English should be substantially improved by a native speaker service or by some professionals. the following is some examples,

in abstract, Digital economy empowering industrial structure transformation is crucial in achieving regional high-quality development. This sentence can be changed into, The attainment of regional high-quality development necessitates the critical role of the digital economy in facilitating the transformation of industrial structures. or The restructuring of the industrial structure, enabled by the digital economy, is essential to attaining high-quality growth in the area.

(7) the authors use mediating model and also use threshold effect model. I suggest you use one of them to make the paper much clearer to discuss one topic and also make the paper shorter and readable. No one can talk all the story in one paper. There are ten tables in the text, so many.

(8) Section 6 is not good. I can translate your text into 'our research method (sample) has problem; we can also add spillover effect but we refuse to do so.' All these discussions are not good to accept this paper. I suggest you think about new directions based on this paper, and combine this section within conclusion part Section 5. do not write a new part.

Reviewer #2: （1）The introduction of research background and significance is not clear.

（2）The author should summarize the existing research gaps and highlight the innovation of this paper after completing the literature review.

（3）The author needs to introduce the reasons for the research method and combine it with the innovation of the research in this paper.

（4）This article has obtained some interesting findings through the models, but these findings need to be further verified from theory or actual conditions. Also, further highlight the contribution of this article.

（5）The discussion should focus on describing the differences between the article study and other scholars' studies, thus highlighting the relevance and academic value of the article, the following literature should be helpful for your research：(1) Reduction pathways identification of Agricultural Water Pollution in Hubei Province, China. (2) A differential game of water pollution management in the trans-jurisdictional river basin

6. PLOS authors have the option to publish the peer review history of their article (what does this mean?). If published, this will include your full peer review and any attached files.

Reviewer #1: No

Reviewer #2: No

---

## [Author Response · Author response to Decision Letter 0]

23 Feb 2024

We feel great thanks for your professional review work on our article. As you are concerned, there are several problems that need to be addressed. According to your nice suggestions, we have made extensive corrections to our previous draft, the detailed corrections are listed in the Word file.

---

## [Decision Letter · Decision Letter 1]

6 Mar 2024

数字经济、创新要素配置与产业结构转型——以中国长三角城市群为例

PONE-D-23-27148R1

Dear Dr. Yang,

We’re pleased to inform you that your manuscript has been judged scientifically suitable for publication and will be formally accepted for publication once it meets all outstanding technical requirements.

Kind regards,

Jianhua Zhu

Academic Editor

PLOS ONE

Additional Editor Comments (optional):

Reviewers' comments:

Reviewer's Responses to Questions

**Comments to the Author**

1. If the authors have adequately addressed your comments raised in a previous round of review and you feel that this manuscript is now acceptable for publication, you may indicate that here to bypass the “Comments to the Author” section, enter your conflict of interest statement in the “Confidential to Editor” section, and submit your "Accept" recommendation.

Reviewer #1: All comments have been addressed

Reviewer #2: (No Response)

2. Is the manuscript technically sound, and do the data support the conclusions?

Reviewer #1: Yes

Reviewer #2: (No Response)

3. Has the statistical analysis been performed appropriately and rigorously? 

Reviewer #1: Yes

Reviewer #2: (No Response)

4. Have the authors made all data underlying the findings in their manuscript fully available?

Reviewer #1: Yes

Reviewer #2: (No Response)

5. Is the manuscript presented in an intelligible fashion and written in standard English?

Reviewer #1: Yes

Reviewer #2: (No Response)

6. Review Comments to the Author

Reviewer #1: (No Response)

Reviewer #2: (No Response)

7. PLOS authors have the option to publish the peer review history of their article (what does this mean?). If published, this will include your full peer review and any attached files.

Reviewer #1: No

Reviewer #2: No

---

## [Editor Report · Acceptance letter]

23 Mar 2024

PONE-D-23-27148R1 

PLOS ONE

Dear Dr. Yang, 

I'm pleased to inform you that your manuscript has been deemed suitable for publication in PLOS ONE. Congratulations! Your manuscript is now being handed over to our production team.

Kind regards, 

on behalf of

Dr. Jianhua Zhu 

Academic Editor

PLOS ONE